# Burden and determinants of self-reported high blood pressure among women of reproductive age in Tanzania: Evidence from 2022 Tanzania demographic and health survey

**Nelson Musilanga** [ORCID]*, **Hussein Nasib, Ambokile Mwakibolwa, Given Jackson, Clarkson Nhanga, Keneth Kijusya**

Department of Internal Medicine, Maranatha Hospital, Mbeya, Tanzania

* nmusilanga@gmail.com

## Abstract

### Background

High blood pressure, commonly referred to as hypertension, remains a prevalent global health concern characterized by elevated arterial pressure, posing significant risks such as cardiovascular diseases, stroke, and kidney diseases. Therefore, this study aimed to assess the burden and determinants of self-reported high blood pressure among women of reproductive age in Tanzania.

### Methods

We utilized population-based cross-sectional data obtained from the Tanzania Demographic and Health Survey (TDHS) 2022. The analysis involved a weighted sample of 15,254 women aged 15–49 years. Multivariable logistic regression models were employed to examine the independent variables associated with self-reported high blood pressure, and the results were presented as adjusted odds ratios (aOR) with a 95% confidence interval (CI). The significance level was set at $p < 0.05$ for all analyses.

### Results

Overall, the mean age of study participants was 29.3 ± 9.8 years, with a self-reported high blood pressure burden of 6.6% among women of reproductive age in Tanzania. Moreover, increased age correlated with higher odds of high blood pressure compared to women aged 15–19 years. Those with a secondary level of education exhibited a higher likelihood of high blood pressure in contrast to women with no education. Married and widowed individuals were more prone to high blood pressure than those who were never married. Additionally, women in higher wealth groups showed a significantly elevated risk of high blood pressure compared to those in the poorest wealth group. Conversely, self-reported poor health status and recent visits to health facilities were associated with significantly higher odds of high blood pressure.

**Data availability statement:** The data is available at Figshare, DOI: https://doi.org/10.6084/m9.figshare.27880485.v1

**Funding:** The author(s) received no specific funding for this work.;

**Competing interests:** The authors have declared that no competing interests exist.

## Conclusion

This study highlights the burden of high blood pressure among reproductive-age women, urging heightened awareness and proactive screening measures. These findings prompt targeted interventions, emphasizing the need for collaborative efforts among stakeholders to effectively curb this health burden.

## Introduction

High blood pressure (BP) is an important modifiable risk factor contributing to heart failure, valvular heart diseases, atrial fibrillation, chronic kidney disease, coronary heart disease, and stroke [1]. High BP or hypertension is defined as a systolic BP greater or equal to 140 mmHg and/or diastolic BP greater or equal to 90 mmHg or a self-reported case under medication [2]. By 2025, the global burden of high BP is projected to reach1.56 billion; two-thirds of this burden is anticipated in low- and middle-income countries [3,4]. In the sub-Saharan Africa region, non-communicable diseases (NCDs), including high BP, are projected to emerge as the leading causes of morbidity and mortality by 2030 [5,6]. According to the World Health Organization (WHO) cardiovascular diseases (CVD) account for 17.9 million deaths annually, making them the most prevalent among NCDs [7].

The development of high BP is influenced by both modifiable and non-modifiable risk factors, such as increasing age, gender, genetic predisposition, unhealthy eating habits, physical inactivity, cigarette smoking, excessive alcohol consumption, and obesity [8]. Furthermore, a preponderance of previous research has also reported that respondent's level of education [9,10], place of residence [11], occupation [12], lack of health insurance coverage [13], and presence of other comorbidities [11], play pivotal roles in the development of high BP.

While high blood pressure is often more prevalent among older populations, recent studies have highlighted an increasing trend of hypertension among younger populations, including women of reproductive age. An analysis of data from the National Health and Nutrition Examination Survey in the United States found that among non-pregnant women aged 20–44 years, the prevalence of hypertension was 16.5% [14]. This trend is particularly concerning in low- and middle-income countries, where lifestyle changes such as increased urbanization, dietary shifts, and reduced physical activity have contributed to rising rates of non-communicable diseases. For instance, a population-based study in Kenya reported a hypertension prevalence of 9.38% among women [15]. Similarly, a prevalence of 9.9% was observed among women of reproductive age in Benin [16]. Furthermore, a study in South Africa supported these findings, showing a prevalence of 22.9% among women aged 20–44 years [17]

According to the WHO, the prevalence of hypertension in Tanzania stands at 34% among adults aged 30–79 years [18]. Although blood pressure measurement is generally available at most primary healthcare facilities, a large portion of individuals with hypertension remain undiagnosed or inadequately treated, highlighting the need for improved public health interventions. Furthermore, a prospective multicenter longitudinal study conducted in district and regional hospitals in Tanzania reported that over 60% of patients experiencing hypertensive crises require inpatient management, underscoring the severity of the condition [19].

Despite several limitations, such as poor patient-clinician communication, self-diagnosis in the absence of an adequate explanation for symptoms, or issues related to health literacy, self-reported medical history data are frequently employed in epidemiological studies to evaluate the prevalence of medical conditions in the general population. Self-reporting of specific medical conditions is a simple and cost-effective method that relies on individuals willingly disclosing their behaviors, beliefs, attitudes, or intentions regarding a particular condition.

Therefore, due to its simplicity and low cost in obtaining health information from individuals, self-reporting has become an integral approach for continuous population monitoring, informing policies aimed at reducing the burden of various diseases [20].

High blood pressure during reproductive age is an important public health issue because it poses significant risks for maternal and neonatal outcomes, including pre-eclampsia and eclampsia. In light of this, while previous studies have explored hypertension in various populations, limited research has focused specifically on women of reproductive age in Tanzania. The current study aims to determine the burden and determinants of self-reported high blood pressure among women of reproductive age in Tanzania, using recent, nationally representative data, with the goal of facilitating early intervention and reducing the burden of hypertensive disorders in pregnancy and associated complications.

## Methods

### Study design, period and setting

This was a population-based cross-sectional study design conducted in Tanzania located in the Eastern Africa from 24th February 2022 to 21st July 2022. Administratively, Tanzania is divided into 3 regions including the islands of Zanzibar. In turn, each administrative region is sub-divided into districts, each district into wards and each ward into villages (lowest administrative units of a country). Based on the 2022 population and housing census, Tanzania has a population of over 61.7 million people, of which 51.3% comprises females [21].

### Data sources

This secondary data analysis was based on the 2022 Tanzania Demographic and Health Surveys (TDHS). This is a nationally representative surveys conducted every four years by the National Bureau of Statistics (NBS) and Zanzibar Bureau of Statistics (ZBS) with financial and technical assistance by ICF International provisioned through the United States Agency for International Development (USAID)-funded Monitoring and Evaluation to Assess and Use Results (MEASURE) DHS program in collaboration with other donor agencies. The 2022 TDHS dataset is publicly available and can be accessed through the DHS program database at http://dhsprogram.com/data/available-datasets.cfm.

### Sampling procedure

The 2022 TDHS sample involved a multi-stage stratified cluster sampling design based on a list of enumeration areas (EAs) delineated for the 2012 Tanzania Population and Housing Census (2012 PHC). The EAs were selected with a probability proportional to their size within each sampling stratum. A total of 629 clusters were selected. Among the 629 EAs, 211 were from urban areas and 418 were from rural areas. Then, 26 households were selected systematically from each cluster, for a total anticipated sample size of 16,354 households for the 2022 TDHS. The detailed sampling procedure has been reported previously [22].

### Study population and sample size

All women of reproductive age, 15 to 49 years, were included in this study. The individual questionnaire of the 2022 TDHS dataset was used to extract the study-specific outcome and explanatory variables from a nationally representative sample of 15,254 women, yielding a response rate of 97%. Given this large sample size, combined with high response rate, provides the study with strong statistical power, well above the standard threshold of 80%. This allows for the detection of even small effect sizes in the associations between risk factors and high blood pressure, despite the relatively low expected prevalence of hypertension in this

population. This level of power ensures that the findings of this study can reliably inform public health interventions.

## Data collection

Data collection was carried out from February 24, 2022 to July 21, 2022 by 18 field teams, 3 teams for Zanzibar and 15 teams for Tanzania Mainland. During fieldwork, enumeration area maps, listing forms, and local leaders assisted the trained field staff in identifying the sampled clusters and households. The team leaders and computer-assisted personal interviewing supervisors were responsible for data quality in the field.

## Outcome variable

The main outcome variable in this study was self-reported high blood pressure. Participants were asked if they had ever been told they have high blood pressure, eliciting a dichotomized response of 'Yes' or 'No.'

## Independent variable

Demographic and socioeconomic variables, such as the respondent's age, place of residence, highest education level, wealth index status, marital status, number of children ever born, working status, coverage by health insurance, self-reported health status, visiting health facility in the last 12 months, and cigarette use, were included as independent variables. The respondent's age was grouped into 5-year age groups. Place of residence was categorized as rural or urban. The highest level of education was separated into four categories: no education, primary level, secondary level, and higher education. The combined wealth index status was categorized into five groups: poorest, poorer, middle, richer, and richest. Marital status was also classified into six groups: never married, married, living with a partner, widowed, divorced, and separated. The total number of children ever born was categorized into 0-children, 1–5 children, and >5 children. Other variables such as the respondent's working status, coverage by health insurance, self-reported health status, visiting a health facility in the last 12 months, and cigarette smoking were dichotomized into 'Yes' or 'No'. Our choice of these independent variables was influenced by variables included in the DHS datasets and previous studies that found these variables to be important factors influencing high blood pressure [8–13].

## Statistical analysis

Analysis was conducted using both descriptive and inferential statistical methods. The descriptive statistics are expressed as numbers with percentages in the form of tables. For inferential statistics, both bivariate and multivariate analyses were applied. The bivariate analysis was performed using the chi-square test to assess the association between independent variables and the outcome variable. Only variables that showed statistical significance ($p < 0.05$) in the bivariate analysis were included in the multivariate analysis. The multivariate analysis was performed using binary logistic regression to investigate the relationship between the independent variables and the outcome variable, while controlling for confounders. Adjusted odds ratios (aOR) were generated to assess the strength of these associations, with 95% confidence intervals (CIs) used for significance testing. The level of significance was set at $p < 0.05$ (2-tailed) for all analyses. Sample weighting was applied to adjust for the cluster sampling design and sampling probabilities across clusters and strata. All analyses were conducted using Statistical Product and Service Solutions (SPSS) for Macintosh, version 25 (IBM Inc., Chicago, Ill., USA).

### Ethical approval and consent to participate

The protocols and data collection procedures received approval from relevant authorities in both Tanzania mainland and Zanzibar, including the National Institute of Medical Research (NIMR), Zanzibar Medical Research Ethical Committee (ZAMREC), the Institutional Review Board of ICF International, and the Centers for Disease Control and Prevention in Atlanta. All participants were asked to provide verbal informed consent after the consent statement was read to them, emphasizing the voluntary nature of the survey. Interviews were conducted under the private conditions afforded by the environments encountered. Confidentiality was ensured by making sure that names of respondents were not written in the data collection tools, and hence, the responses remained anonymous. The approval to use the data in this study was obtained from MEASURE DHS with authorization number 194069.

## Results

### Sociodemographic characteristics of the study participants

As shown in Table 1, a total of 15,254 women of reproductive age from all regions of Tanzania were included in the study. The mean age of study participants was $29.3 \pm 9.8$, ranging from 15 to 49 years old. Participants exhibited a range of educational backgrounds, with 48.6% predominantly holding a primary education level, while 1.4% possessed higher education qualifications. A majority were married (44.3%), and nearly two-thirds of participants (64.3%) were living in rural areas. Furthermore, participants represented diverse economic profiles, with the poorest accounting for 14.9% and the richest comprising 26.5%.

Regarding health and lifestyle factors, a significant portion of participants (94.3%) were not covered by health insurance, 99.5% had not smoked cigarettes, and approximately 53.8% reported visiting a health facility in the past 12 months. Additionally, 52.7% reported having a good health status.

### Burden and determinants of self-reported high blood pressure

Overall, 1,014 (6.6%) of the study participants self-reported having high blood pressure, as indicated in Table 2. Comparatively, individuals with high blood pressure exhibited a higher proportion among those aged ≥40 years (40.5% vs. 18.1%, $p < 0.001$) than participants without high blood pressure. The group with high blood pressure also showed a higher representation of individuals in the richer and richest wealth status categories (70.6% vs. 47.1%, $p < 0.001$). Additionally, a significantly proportion of women who had given birth to more than three children were observed in the high blood pressure group (45.3% vs. 30.6%, $p < 0.01$). However, percentages of cigarette smoking and having given birth more than twice in the past 5 years were comparable between the groups (0.5% vs. 0.4% and 2.0% vs. 2.4%, respectively).

Furthermore, the multivariate logistic regression model was fitted with baseline covariates associated with self-reported high blood pressure determined through bivariate analysis at a significance level of $p < 0.05$, as indicated in Table 2. Several factors were independently associated with self-reported high blood pressure among women of reproductive age in Tanzania. These factors include age-groups: 20–24 (aOR: 2.45, 95% CI: 1.64–3.65), 25–29 (aOR: 3.77, 95% CI: 2.48–5.71), 30–34 (aOR: 4.60, 95% CI: 2.99–7.07), 35–39 (aOR: 6.26, 95% CI: 4.04–9.70), 40–44 (aOR: 7.62, 95% CI: 4.86–11.95), and 45–49 (aOR: 11.78, 95% CI: 7.45–18.62); secondary level of education (aOR: 1.76, 95% CI: 1.36–2.28); marital status-married (aOR: 1.57, 95% CI: 1.19–2.08) and widowed (aOR: 1.54, 95% CI: 1.11–2.12); middle wealth status (aOR: 2.27, 95% CI: 1.64–3.16); richer wealth status (aOR: 2.97, 95% CI: 2.13–4.14); richest wealth status (aOR: 5.06, 95% CI: 3.58–7.14); having a self-reported poor health status (aOR: 2.86, 95%

**Table 1.  Sociodemographic characteristics of the study participants (N = 15254).**

| | | | Self-Reported High BP | |
|---|---|---|---|---|
| **Variables** | **Categories** | **Total N = 15254** | **Yes (n = 1014)** | **No (n = 14240)** |
| **Age (Years)** | 15–19 | 3142 (20.6%) | 41 (4.0%) | 3101 (21.8%) |
| | 20–24 | 2710 (17.8%) | 94 (9.3%) | 2616 (18.4%) |
| | 25–29 | 2500 (16.4%) | 146 (14.4%) | 2354 (16.5%) |
| | 30–34 | 2041 (13.4%) | 148 (14.6%) | 1893 (13.3%) |
| | 35–39 | 1882 (12.3%) | 175 (17.3%) | 1707 (12.0%) |
| | 40–44 | 1550 (10.2%) | 174 (17.2%) | 1376 (9.7%) |
| | 45–49 | 1429 (9.4%) | 236 (23.3%) | 1193 (8.4%) |
| **Education level** | No education | 2387 (15.6%) | 106 (10.5%) | 2281 (16.0%) |
| | Primary | 7413 (48.6%) | 475 (46.8%) | 6938 (48.7%) |
| | Secondary | 5235 (34.3%) | 409 (40.3%) | 4826 (33.9%) |
| | Higher | 219 (1.4%) | 24 (2.4%) | 195 (1.4%) |
| **Marital status** | Never married | 4232 (27.7%) | 121 (11.9%) | 4111 (28.9%) |
| | Married | 6751 (44.3%) | 624 (61.5%) | 6127 (43.0%) |
| | Cohabiting | 2400 (15.7%) | 100 (9.9%) | 2300 (16.2%) |
| | Divorced/Separated | 1514 (9.9%) | 132 (13.0%) | 1382 (9.7%) |
| | Widowed | 357 (2.3%) | 37 (3.6%) | 320 (2.2%) |
| **Residence** | Urban | 5441 (35.7%) | 502 (49.5%) | 4939 (34.7%) |
| | Rural | 9813 (64.3%) | 512 (50.5%) | 9301 (65.3%) |
| **Zone** | | | | |
| | Western | 1127 (7.4%) | 39 (3.8%) | 1088 (7.6%) |
| | Northern | 1461 (9.6%) | 103 (10.3%) | 1358 (9.5%) |
| | Central | 1328 (8.7%) | 37 (3.6%) | 1291 (9.1%) |
| | Southern Highlands | 1209 (7.9%) | 74 (7.3%) | 1135 (8.0%) |
| | Southern | 794 (5.2%) | 22 (2.2%) | 772 (5.4%) |
| | South West Highlands | 1767 (11.6%) | 51 (5.0%) | 1716 (12.1%) |
| | Lake | 3148 (20.6%) | 112 (11.0%) | 3036 (21.3%) |
| | Eastern | 1852 (12.1%) | 205 (20.2%) | 1647 (11.6%) |
| | Zanzibar | 2568 (16.8%) | 371 (36.6%) | 2197 (15.4%) |
| **Wealth status** | Poorest | 2271 (14.9%) | 51 (5.0%) | 2220 (15.6%) |
| | Poorer | 2498 (16.4%) | 77 (7.6%) | 2421 (17.0%) |
| | Middle | 3063 (20.1%) | 170 (16.8%) | 2893 (20.3%) |
| | Richer | 3378 (22.1%) | 241 (23.8%) | 3137 (22.0%) |
| | Richest | 4044 (26.5%) | 475 (46.8%) | 3569 (25.1%) |
| **Smoke cigarette** | Yes | 69 (0.5%) | 5 (0.5%) | 64 (0.4%) |
| | No | 15185 (99.5%) | 1009 (99.5%) | 14176 (99.6%) |
| **Currently working** | Yes | 8923 (58.5%) | 679 (67.0%) | 8244 (57.9%) |
| | No | 6331 (41.5%) | 335 (33.0%) | 5996 (42.1%) |
| **Covered by health insurance** | Yes | 868 (5.7%) | 98 (9.7%) | 770 (5.4%) |
| | No | 14386 (94.3%) | 916 (90.3%) | 13470 (94.6%) |
| **Self-Reported Health Status** | Very good | 3105 (20.4%) | 214 (21.1%) | 2891 (20.3%) |
| | Good | 8042 (52.7%) | 496 (48.9%) | 7546 (53.0%) |
| | Moderate | 3956 (25.9%) | 278 (27.4%) | 3678 (25.8%) |
| | Poor | 151 (1.0%) | 26 (2.6%) | 125 (0.9%) |
| **Visited HF in the Last 12 months** | Yes | 8207 (53.8%) | 661 (65.2%) | 7546 (53.0%) |
| | No | 7047 (46.2%) | 353 (34.8%) | 6694 (47.0%) |
| **Total Children Ever Born** | No Child | 4138 (27.1%) | 121 (11.9%) | 4017 (28.2%) |

*(Continued)*

**Table 1.** (Continued)

| | | | Self-Reported High BP | |
|---|---|---|---|---|
| Variables | Categories | Total N = 15254 | Yes (n = 1014) | No (n = 14240) |
| | 1 to 3 Children | 6300 (41.3%) | 434 (42.8%) | 5866 (41.2%) |
| | More than 3 Children | 4816 (31.6%) | 459 (45.3%) | 4357 (30.6%) |
| Birth in Last 5 Years | No Birth | 7573 (49.6%) | 554 (54.6%) | 7019 (49.3%) |
| | Once or Twice | 7324 (48.0%) | 440 (43.4%) | 6884 (48.3%) |
| | More than Twice | 357 (2.3%) | 20 (2.0%) | 337 (2.4%) |

CI: 1.77–4.63); visiting a health facility in the past 12 months (aOR: 1.38, 95% CI: 1.20–1.60); and giving birth once or twice in the last 5 years (aOR: 0.80, 95% CI: 0.67–0.96).

## Discussion

The study found a 6.6% prevalence of high blood pressure among women of reproductive age. Several factors independently correlated with self-reported high blood pressure, including older age-groups, secondary education level, marital status (being married or widowed), middle, richer, and richest wealth statuses, self-reported poor health status, recent health facility visits, and having given birth once or twice in the last 5 years.

Our findings are consistent with a prior report from a community-based survey in rural Mwanza [23], which recorded a relatively similar prevalence rate of 8%. However, our findings were notably lower compared to several other community-based studies across the country. For instance, studies in Northern Tanzania reported a prevalence of 28.0% [24], while another in rural Morogoro noted 29.3% [25], and the national representative STEPS survey in 2012 found a prevalence of 25.9% [26]. Mwanri et al. examined 920 pregnant women aged at least 20 years attending antenatal clinics in Tanzania and found that 6.9% had high blood pressure (HBP), with a higher prevalence in urban areas (8.1%) compared to rural areas (4.4%) [27]. The lower prevalence in our study might be due to the inclusion of younger participants, as 38.9% were aged 15–24 years, contrasting previous studies that involved older participants, where high blood pressure is more common. Additionally, it is important to note that many individuals with hypertension in our study might not be aware of their condition [25].

When we compare our findings with reports from other countries, we uncover a wide range of outcomes. A cross-sectional survey conducted in Nakaseke health district in Uganda reported a quite similar prevalence rate of self-reported high blood pressure at 6.3% [28]. Population-based studies among women in Kenya and Benin reported prevalence rates of 9.38% and 9.9%, respectively [29,15]. Additionally, other studies indicated higher prevalence rates of high blood pressure than those observed in our study. For instance, a study in Northern China reported a prevalence of 19.0% [30], South Africa reported a prevalence of 23.6% [17], while Brazil showed 24.1% [9], and the overall age-standardized rate among women in the United States reached 27.0% [31]. These differences underscore significant disparities in the prevalence of self-reported high blood pressure among diverse populations, potentially influenced by factors such as methodological differences, population healthcare-seeking behavior, socioeconomic status, sociocultural norms, and awareness within a population regarding specific health conditions.

In this study, the likelihood of self-reported high blood pressure increased with age, ranging from 14.4% among individuals aged 25–29 to 23.3% among those aged 44–49 years, aligning with previous reports [32,33]. Age-related elevation in high blood pressure has been associated with arterial stiffness and reduced vessel elasticity due to arteriosclerotic structural

**Table 2. Bivariate and multivariate analysis for factors associated with self-reported high blood pressure.**

| Variables | Categories | Self-Reported High BP | | Crude | | Adjusted | |
|---|---|---|---|---|---|---|---|
| | | Yes n (%) | No n (%) | OR (95% CI) | P-value | OR (95% CI) | P-value |
| **Age (Years)** | 15–19 | 41 (4.0%) | 3101 (21.8%) | 1 (Reference) | | 1 (Reference) | |
| | 20–24 | 94 (9.3%) | 2616 (18.4%) | 2.72 (1.88–3.94) | p < 0.001 | 2.45 (1.64–3.65) | **p < 0.001** |
| | 25–29 | 146 (14.4%) | 2354 (16.5%) | 4.69 (3.30–6.66) | p < 0.001 | 3.77 (2.48–5.71) | **p < 0.001** |
| | 30–34 | 148 (14.6%) | 1893 (13.3%) | 5.91 (4.17–8.40) | p < 0.001 | 4.60 (2.99–7.07) | **p < 0.001** |
| | 35–39 | 175 (17.3%) | 1707 (12.0%) | 7.75 (5.49–10.95) | p < 0.001 | 6.26 (4.04–9.70) | **p < 0.001** |
| | 40–44 | 174 (17.2%) | 1376 (9.7%) | 9.56 (6.77–13.52) | p < 0.001 | 7.62 (4.86–11.95) | **p < 0.001** |
| | 45–49 | 236 (23.3%) | 1193 (8.4%) | 14.96 (10.67–20.99) | p < 0.001 | 11.78 (7.45–18.62) | **p < 0.001** |
| **Education level** | No education | 106 (10.5%) | 2281 (16.0%) | 1 (Reference) | | 1 (Reference) | |
| | Primary | 475 (46.8%) | 6938 (48.7%) | 1.47 (1.19–1.83) | p < 0.001 | 1.17 (0.93–1.48) | p = 0.175 |
| | Secondary | 409 (40.3%) | 4826 (33.9%) | 1.82 (1.47–2.27) | p < 0.001 | 1.76 (1.36–2.28) | **p < 0.001** |
| | Higher | 24 (2.4%) | 195 (1.4%) | 2.65 (1.66–4.22) | p < 0.001 | 1.47 (0.87–2.47) | p = 0.147 |
| **Marital status** | Never married | 121 (11.9%) | 4111 (28.9%) | 1 (Reference) | | 1 (Reference) | |
| | Married | 624 (61.5%) | 6127 (43.0%) | 3.46 (2.84–4.22) | p < 0.001 | 1.57 (1.19–2.08) | **p = 0.002** |
| | Cohabiting | 100 (9.9%) | 2300 (16.2%) | 1.48 (1.13–1.94) | p = 0.005 | 0.99 (0.71–1.38) | p = 0.932 |
| | Divorced/Separated | 132 (13.0%) | 1382 (9.7%) | 3.93 (2.67–5.78) | p < 0.001 | 1.40 (0.89–2.18) | p = 0.145 |
| | Widowed | 37 (3.6%) | 320 (2.2%) | 3.25 (2.52–4.18) | p < 0.001 | 1.54 (1.11–2.12) | **p = 0.009** |
| **Residence** | Urban | 502 (49.5%) | 4939 (34.7%) | 1 (Reference) | | 1 (Reference) | |
| | Rural | 512 (50.5%) | 9301 (65.3%) | 0.54 (0.48–0.62) | p < 0.001 | 1.02 (0.86–1.20) | p = 0.852 |
| **Wealth status** | Poorest | 51 (5.0%) | 2220 (15.6%) | 1 (Reference) | | – | |
| | Poorer | 77 (7.6%) | 2421 (17.0%) | 1.38 (0.97–1.98) | p = 0.075 | 1.31 (0.91–1.88) | p = 0.151 |
| | Middle | 170 (16.8%) | 2893 (20.3%) | 2.56 (1.86–3.52) | p < 0.001 | 2.27 (1.64–3.16) | **p < 0.001** |
| | Richer | 241 (23.8%) | 3137 (22.0%) | 3.34 (2.46–4.55) | p < 0.001 | 2.97 (2.13–4.14) | **p < 0.001** |
| | Richest | 475 (46.8%) | 3569 (25.1%) | 5.79 (4.32–7.77) | p < 0.001 | 5.06 (3.58–7.14) | **p < 0.001** |
| **Smoke cigarette** | Yes | 5 (0.5%) | 64 (0.4%) | 1.09 (0.44–2.73) | p = 0.841 | – | – |
| | No | 1009 (99.5%) | 14176 (99.6%) | 1 (Reference) | | – | |
| **Currently working** | Yes | 679 (67.0%) | 8244 (57.9%) | 1.47 (1.29–1.69) | p < 0.001 | 0.88 (0.76–1.02) | p = 0.099 |
| | No | 335 (33.0%) | 5996 (42.1%) | 1 (Reference) | | 1 (Reference) | |
| **Covered by health insurance** | Yes | 98 (9.7%) | 770 (5.4%) | 1.87 (1.50–2.33) | p < 0.001 | 0.91 (0.71–1.16) | p = 0.437 |
| | No | 916 (90.3%) | 13470 (94.6%) | 1 (Reference) | | 1 (Reference) | |
| **Self-Reported Health Status** | Very good | 214 (21.1%) | 2891 (20.3%) | 1 (Reference) | | 1 (Reference) | |
| | Good | 496 (48.9%) | 7546 (53.0%) | 0.89 (0.75–1.05) | p = 0.160 | 0.94 (0.79–1.12) | p = 0.506 |
| | Moderate | 278 (27.4%) | 3678 (25.8%) | 1.02 (0.85–1.23) | p = 0.825 | 1.06 (0.87–1.29) | p = 0.562 |
| | Poor | 26 (2.5%) | 125 (0.9%) | 2.81 (1.80–4.38) | p < 0.001 | 2.86 (1.77–4.63) | **p < 0.001** |
| **Visited HF in the Last 12 months** | Yes | 661 (65.2%) | 7546 (53.0%) | 1.66 (1.45–1.90) | p < 0.001 | 1.38 (1.20–1.60) | **p < 0.001** |
| | No | 353 (34.8%) | 6694 (47.0%) | 1 (Reference) | | 1 (Reference) | |
| **Total Children Ever Born** | No Child | 121 (11.9%) | 4017 (28.2%) | 1 (Reference) | | 1 (Reference) | |
| | 1 to 3 Children | 434 (42.8%) | 5866 (41.2%) | 2.46 (2.00–3.02) | p < 0.001 | 1.13 (0.84–1.54) | p = 0.419 |
| | >3 Children | 459 (45.3%) | 4357 (30.6%) | 3.50 (2.85–4.29) | p < 0.001 | 1.32 (0.93–1.88) | p = 0.118 |
| **Birth in Last 5 Years** | No Birth | 554 (54.6%) | 7019 (49.3%) | 1 (Reference) | | 1 (Reference) | |
| | Once or Twice | 440 (43.4%) | 6884 (48.3%) | 0.81 (0.71–0.92) | p = 0.001 | 0.80 (0.67–0.96) | **p = 0.014** |
| | More than Twice | 20 (2.0%) | 337 (2.4%) | 0.75 (0.48–1.19) | p = 0.224 | 0.88 (0.53–1.45) | p = 0.609 |

changes and calcification [34]. Women with secondary level of educations were more likely to report high blood pressure compared to those with no education. However, those with higher education were less likely to have high blood pressure 2.4%. Individuals with higher levels

of education often exhibit a greater awareness of health-related behaviors, including regular exercise, balanced diets, and stress management [35].

The loss of a spouse through widowhood can have adverse effects on a woman's health, including an increased risk of high blood pressure as observed in our study. The emotional stress, social isolation, and potential lifestyle changes following the loss of a partner can contribute to higher blood pressure levels [36]. Conversely, the association between being married and self-reported high blood pressure in our setting may be influenced by a significant number of participants, the majority of whom were married (44.3%).

Furthermore, richest wealth status had a 5-fold increased risk of high blood pressure among women. This is contrast with reports from previous studies [37,38]. However, increased wealth might afford access to certain lifestyle choices that can negatively impact blood pressure, such as excessive alcohol consumption, high-calorie diets, or a lack of emphasis on regular exercise [39].

A self-reported poor health status was linked to more than twice the increased risk of high blood pressure compared to individuals reporting very good health status. Poor health status leads to physiological changes impacting the cardiovascular system, heightening the risk of high blood pressure. Additionally, women who had visited a health facility in the past 12 months were more likely to report high blood pressure than those who had not. This may be attributed to the incomplete functioning of community-level screening, consequently resulting in a higher detection rate at the health facility level. These findings highlight the need for healthcare providers to implement community-level screening for high blood pressure.

Moreover, the decreased likelihood of high blood pressure among women who have given birth once or twice in the past 5 years aligns with the recommendation to wait at least 18 to 24 months after a live birth before attempting another pregnancy [40]. Multiple pregnancies with inadequate spacing may heighten the risk of gestational hypertension or preeclampsia due to physiological changes during pregnancy, including elevated blood volume and hormonal fluctuations. These factors can potentially predispose women to chronic high blood pressure later in life.

The findings of this study are particularly useful, especially in rural Tanzania, where limited access to antenatal care services significantly raises pregnancy-related hypertension risks. Women in these areas face barriers such as long distances to health facilities, lack of transportation, financial constraints, and insufficient healthcare workers [41]. These challenges lead to reduced ANC visits, limiting early detection and management of hypertension. Pregnancy-induced hypertension, like pre-eclampsia, can result in severe complications, including maternal and neonatal mortality, exacerbating the already high maternal mortality rate of 104 deaths per 100,000 live births as of 2022. Therefore, the study highlights the urgent need to improve antenatal care access in Tanzania, particularly in under-resourced regions, which could help mitigate risks and contribute to achieving Sustainable Development Goal (SDG) 3.1, aiming to reduce global maternal mortality to less than 70 per 100,000 live births by 2030.

## Strength and limitation

Our study has several strengths. It utilizes a large sample size of 15,254 women of reproductive age, enhancing the reliability of the findings within this demographic. Furthermore, the study design allows for valuable understanding of high blood pressure prevalence among women in Tanzania. However, there are important limitations to consider. The cross-sectional design restricts our ability to establish cause-and-effect relationships among the identified variables; therefore, these findings should be interpreted cautiously. The reliance on self-reported high blood pressure may introduce recall bias, potentially leading to misreporting or overreporting of prevalence rates. Additionally, the sample includes only women aged 15 to 49 years, with a

mean age of 29.3 years. This demographic focus may not fully represent the broader Tanzanian population, particularly since high blood pressure is more prevalent in older individuals (e.g., those aged 50 years and above). The lack of data on body mass index (BMI), dietary habits, daily physical activity, and chronic diseases such as diabetes limits our understanding of other potential risk factors associated with hypertension. Lastly, since the sample consists solely of women, this study does not account for potential gender differences in hypertension prevalence, which may further limit the generalizability of the findings.

## Conclusion

The culmination of this study underscores several significant insights, shaping our understanding and potentially guiding interventions in this demographic. Firstly, the prevalence rates observed in this study reflect an alarming trend, highlighting the substantial burden of high blood pressure among women of reproductive age. The figures underscore the imperative need for heightened awareness, proactive screening measures, and targeted interventions aimed at mitigating the risks associated with high blood pressure among women.

Moreover, the determinants identified within this research elucidate the complex interplay of various factors contributing to elevated blood pressure levels. Understanding these determinants provides a framework for developing tailored strategies encompassing lifestyle modifications, education initiatives, and improved healthcare accessibility.

Importantly, the implications drawn from this study reverberate across public health policies, clinical practice, and community-based interventions. Implementing targeted screening programs, fostering health-promoting behaviors, and enhancing healthcare access are pivotal in mitigating the burden of high blood pressure in this demographic. Additionally, an interdisciplinary approach encompassing healthcare providers, public health officials, policymakers, and community stakeholders is imperative to effectuate sustainable and impactful changes

## Supporting information

**S1 File. Used dataset for self-reported high BP,** https://doi.org/10.6084/m9.figshare.27880485.v1.
(SAV)

## Acknowledgement

We are grateful to the Monitoring and Evaluation to Assess and Use Results-Demographic and Health Surveys (MEASURE DHS) and the National Bureau of Statistics (NBS) for providing free access to the original dataset. We also acknowledge that an earlier version of this manuscript has been presented as a preprint on medRxiv. This preprint served as a preliminary dissemination of our research findings.

## Author contributions

**Conceptualization:** Nelson Musilanga.

**Data curation:** Nelson Musilanga, Hussein Nasib, Ambokile Mwakibolwa, Given Jackson.

**Formal analysis:** Nelson Musilanga, Hussein Nasib, Ambokile Mwakibolwa, Given Jackson, Clarkson Nhanga.

**Investigation:** Nelson Musilanga.

**Methodology:** Nelson Musilanga, Hussein Nasib, Ambokile Mwakibolwa, Given Jackson, Clarkson Nhanga, Keneth Kijusya.

**Project administration:** Nelson Musilanga.

**Software:** Nelson Musilanga.

**Supervision:** Nelson Musilanga.

**Writing – original draft:** Nelson Musilanga, Hussein Nasib, Ambokile Mwakibolwa.

**Writing – review & editing:** Nelson Musilanga, Hussein Nasib, Given Jackson, Clarkson Nhanga, Keneth Kijusya.

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
