## [Decision Letter · Decision Letter 0]

2 Oct 2024

PONE-D-24-35959Burden and Determinants of Self-Reported High Blood Pressure Among Women of Reproductive Age In Tanzania: Evidence from 2022 Tanzania Demographic and Health SurveyPLOS ONE

Dear Dr. Musilanga,

Thank you for submitting your manuscript to PLOS ONE. After careful consideration, we feel that it has merit but does not fully meet PLOS ONE’s publication criteria as it currently stands. Therefore, we invite you to submit a revised version of the manuscript that addresses the points raised during the review process.

**Three reports have been obtained. Please find these below.**

We look forward to receiving your revised manuscript.

Kind regards,

Muhammad Haroon Stanikzai

Academic Editor

PLOS ONE

**Journal Requirements:**

2. We note that your Data Availability Statement is currently as follows: All relevant data are within the manuscript and its Supporting Information files

**Additional Editor Comments:**

- In the logistical regression authors compare self-health status very bad to very good when there are only 4 very bad answers which leads to significant uncertainty. It might be useful to combine very bad with bad.

- Please proofread the entire manuscript for language corrections (there are some typos).

Reviewers' comments:

Reviewer's Responses to Questions

**Comments to the Author**

1. Is the manuscript technically sound, and do the data support the conclusions?

Reviewer #1: Yes

Reviewer #2: Yes

Reviewer #3: Yes

2. Has the statistical analysis been performed appropriately and rigorously? 

Reviewer #1: Yes

Reviewer #2: Yes

Reviewer #3: I Don't Know

3. Have the authors made all data underlying the findings in their manuscript fully available?

Reviewer #1: Yes

Reviewer #2: Yes

Reviewer #3: Yes

4. Is the manuscript presented in an intelligible fashion and written in standard English?

Reviewer #1: Yes

Reviewer #2: Yes

Reviewer #3: Yes

5. Review Comments to the Author

**Reviewer #1:**  Dear Editors and Authors,

Thank you for giving me the opportunity to review the manuscript, titled “Burden and Determinants of Self-Reported High Blood Pressure Among Women of Reproductive Age In Tanzania: Evidence from 2022 Tanzania Demographic and Health Survey”.

The authors of the study chose a very important topic related to high blood pressure (HBP) in reproductive age women. The authors used data from over 15,000 women from the Tanzania DHS 2022, and by employing a binary logistic regression model, the authors assessed predictors of self-reported HBP. They found that 6.6% of women, with mean age of 29.3 years, had HBP. In terms of predictors of HBP, older age women had higher BP, women with secondary education had higher BP, married and widow women had higher BP, richer women had higher BP, and women with self-reported poor health and recent visits to health facilities had higher BP.

My assessment of the manuscript is that it is well written and is well organized. The findings from this study have the potential to impact health interventions and policy to improve clinical guidelines on BP management and control, and to improve women’s health as a whole. However, the authors need to address the following issues before the paper can be considered for publication.

Abstract

The word “bracket” can be changed to “group” here, and elsewhere in the manuscript.

In the conclusion, it says “This study highlights the substantial burden of …”, and I’m not sure whether the 6.6% burden is substantial. The authors may opt to delete the word “substantial”.

Introduction

The introduction section starts well; however, when it narrows down on identifying the research gap, it fails to provide relevant and convincing evidence to justify the study, which is to examine the prevalence and predictors of HBP in women of reproductive age who are mainly young women 15-49 years old. Considering HBP is more prevalent in older people, mainly 50+ years, the authors need to provide strong justification for choosing this study population and examining HBP among them.

Methods

Under “study population and sample size” the authors should provide some narrative on what the study power is, using the sample size of 15254 women.

If there is a question on use of anti-hypertensive medications in the TDHS2022, the authors may use it to define their outcome, in the addition to the question “whether a participant reported having HBP”.

Results

Under “burden and determinants of self-reported HBP” the authors do not need to report p-values for aORs. However, aORs (95%CI) are enough in the narrative, and aORs (95%CI) and p-values can be kept in the tables. In Table 2, the authors should report aORs for “Wealth status”, Smoke cigarette, Self-report health status, Birth in last 5-years, because currently the authors have deleted them.

Discussion

The discussion needs to start from “The study found a 6.6% prevalence of ….”. Please delete the sentences about this, as they are unnecessary and redundant.

Under “strengths and limitations”, please start by stating the strengths of the study first; then write the limitations. One of the limitations that the authors did not discuss is the lack of data on BMI (or other measures of obesity), on dietary habits, diabetes, and on daily exercise. These are important confounders of HBP. Furthermore, the authors stated that this sampled population may genuinely represent the Tanzania population. I do not think this is correct, given the sample covers only women 15-49 years old, with mean age 29.3 years. The sample seems to be from a younger population, and we know that HBP is more prevalent in older people (e.g., 50+ years). In addition, this sample covers only women. There is evidence that sex is associated with HBP. Thus, this sample cannot be a reprehensive sample of the Tanzania population. The authors should acknowledge this as a limitation.

**Reviewer #2: ** Dear Authors,

The manuscript discussed the prevalence and determinants of self-reported high blood pressure among women of reproductive age in Tanzania. The authors provided good information about the prevalence and the determinants of HBP based on the data from a national population-based survey with good number of samples. However, the authors need to revise the manuscript as below points to improve the quality of this paper.

Introduction

1- Please add a paragraph on why it is important to investigate the prevalence of high blood pressure among women of reproductive age. Why you chose this age group? And please provide a literature review of similar studies about prevalence and determinants of high blood pressure about women of reproductive age.

Result

1- In table 1, you can delete the frequency and percent for answer “No” to the self-reported high BP. Presenting data for yes, can show the frequency and percentage for “no” answer as well. I have the same comment for table 2. I recommend you present the bivariate analysis results in table 1 and the results of multivariate regression analysis in table 2.

Discussion

1- I recommend comparing the result of your study with the result of studies that reported the prevalence of high blood pressure in the same age group of female patients in Tanzania. It will show the difference between prevalence based on self-reported HBP and health facility reported HBP. In addition, I recommend discussing more on the trend of HBP prevalence among women till 2030 and connect your recommendations to the Sustainable Development Goals.

**Reviewer #3:**  Dear Editors and Authors,

Thank you for the opportunity to review the manuscript titled "Burden and Determinants of Self-Reported High Blood Pressure Among Women of Reproductive Age in Tanzania: Evidence from the 2022 Tanzania Demographic and Health Survey."

Overall, the manuscript is well-crafted, logically structured, and addresses a significant public health issue. The findings have the potential to inform health interventions, clinical guidelines, and policies focused on blood pressure management to enhance women's health in Tanzania. However, I have made some detailed observations, some of which may require minor revisions to strengthen the paper.

Title and Abstract:

• The title is clear and informative, accurately reflecting the study's focus on high blood pressure among women of reproductive age in Tanzania.

• The abstract provides a comprehensive summary, covering the background, methods, results, and conclusion. It presents the study's importance, data sources, sample size, and key findings effectively.

Introduction:

• The introduction has several strengths, including a clear emphasis on the global significance of hypertension and its impact on women of reproductive age. It effectively outlines key risk factors and highlights the burden of hypertension in low- and middle-income countries.

• To further enhance the introduction, it would be helpful to include information on Tanzania's health indicators, such as the current prevalence of hypertension and the challenges within the healthcare system. Adding a brief overview of Tanzania's current efforts or policies to manage hypertension would also provide important context.

• Concluding the introduction with a statement addressing the knowledge gap would strengthen the argument for the study's importance. For example, "While previous studies have explored hypertension in various populations, limited research has focused specifically on women of reproductive age in Tanzania, using recent, nationally representative data."

• In line 97, there is a small formatting issue as the space is missing between “reach1.56billion”.

Methods

• The study design (population-based cross-sectional) and the use of national survey data (TDHS 2022) are appropriate for the research question.

• Sampling procedures, data collection methods, and the definition of the outcome variable (self-reported high blood pressure) are clearly described.

• Statistical analysis is detailed, including the use of multivariable logistic regression and the presentation of adjusted odds ratios with confidence intervals.

Results:

• There are some sections where sentence structure can be simplified for better clarity and help convey the study’s key findings more effectively.

Discussion

• The discussion clearly explains the findings and compares the findings with those of previous research in Tanzania and other countries. It provides a detailed analysis of factors associated with high blood pressure in the study population.

• However, it would benefit from integrating more context-specific information about Tanzania. For example, including information on how limited access to antenatal care services in rural Tanzania affects pregnancy-related hypertension risks would provide readers with a clearer understanding of the study's outcomes.

• The first two sentences in the discussion are repetitive as they already mentioned in the introduction. It may be helpful to paraphrase these sentences, start with a different sentence, or make them shorter.

6. PLOS authors have the option to publish the peer review history of their article (what does this mean? ). If published, this will include your full peer review and any attached files.

**Do you want your identity to be public for this peer review?** For information about this choice, including consent withdrawal, please see our Privacy Policy .

Reviewer #1: **Yes: ** Dr Essa Tawfiq

Reviewer #2: **Yes: ** Narges Neyazi

Reviewer #3: **Yes: ** Massoma Jafari

---

## [Author Response · Author response to Decision Letter 0]

26 Oct 2024

Dear Editor,

RE: REVISED MANUSCRIPT (ID 1240457) SUBMISSION

We are grateful for the helpful feedback provided by the reviewers, which helped us in improving the quality of the manuscript. We carefully addressed all points raised and subsequently modified the manuscript accordingly. Below, you'll find point-to-point responses to the comments made by you and your reviewers.

Editor’s Comments to the Author

Comment 1: In the logistical regression authors compare self-health status very bad to very good when there are only 4 very bad answers which leads to significant uncertainty. It might be useful to combine very bad with bad

Authors Response: Thank you for your valuable suggestion. We have combined the "very bad" and "bad" categories into a single group labeled "Poor" health status to reduce uncertainty and improve the stability of the logistic regression model. The updated categories are now "Very Good," "Good," "Moderate," and "Poor." We have rerun the analysis, and these changes have been reflected in the revised manuscript accordingly.

Comment 2: Please proofread the entire manuscript for language corrections (there are some typos)

Authors Response: Thank you for bringing this to our attention. We have noted your comment and made the necessary language corrections throughout the manuscript, including addressing typos.

Reviewer #1 Comments to the Author

Comment 1: In the Abstract, the word “bracket” can be changed to “group” here, and elsewhere in the manuscript.

Authors Response: Thank you for the suggestion. We have replaced the word 'bracket' with 'group' in the abstract and throughout the manuscript, as recommended

Comment 2: In the conclusion, it says “This study highlights the substantial burden of …”, and I’m not sure whether the 6.6% burden is substantial. The authors may opt to delete the word “substantial”.

Authors Response: Thank you for your suggestion. We have deleted the word 'substantial' from the conclusion, as recommended.

Comment 3: Provide relevant and convincing evidence to justify the study, which is to examine the prevalence and predictors of HBP in women of reproductive age who are mainly young women 15-49 years old. Considering HBP is more prevalent in older people, mainly 50+ years, the authors need to provide strong justification for choosing this study population and examining HBP among them.

Authors Response:

We appreciate your thoughtful feedback. In the revised manuscript, we have provided relevant evidence from global and regional studies that support the rationale for investigating this age group (Lines 111 to 130 and 142 to 144).

Comment 4: Under “study population and sample size” the authors should provide some narrative on what the study power is, using the sample size of 15254 women.

Authors Response: Thank you for your comment. With a sample size of 15,254 women, the study achieves a high statistical power, well above the conventional threshold of 80%. We have revised the manuscript to include this narrative under the "Study Population and Sample Size" section, which can be found in lines 215-223.

Comment 5: If there is a question on use of anti-hypertensive medications in the TDHS2022, the authors may use it to define their outcome, in the addition to the question “whether a participant reported having HBP.

Authors Response: The TDHS 2022 includes a question on "Currently taking medications to control blood pressure." However, only 1,014 participants (6.6% of the study population) responded to this question, and all who answered "Yes" also reported having high blood pressure. Given this overlap and the low response rate, we decided it was more appropriate to use self-reported high blood pressure alone as the primary outcome measure to ensure consistency and minimize potential bias.

Comment 6: Under “burden and determinants of self-reported HBP” the authors do not need to report p-values for aORs. However, aORs (95%CI) are enough in the narrative, and aORs (95%CI) and p-values can be kept in the tables.

Authors Response: Thank you for the feedback. We have agreed and revised the manuscript to report only aORs (95% CI) in the narrative, while retaining both aORs (95% CI) and p-values in the tables.

Comment 7: In Table 2, the authors should report aORs for “Wealth status”, Smoke cigarette, Self-report health status, Birth in last 5-years, because currently the authors have deleted them.

Authors Response:

Thank you for your observation. In the revised manuscript, we have now included adjusted odds ratios (aORs) for “Wealth status,” “Self-report health status,” and “Birth in the last 5 years” from the multivariate analysis. However, we have excluded “Smoke cigarette” from the final model because it did not show statistical significance in the bivariate logistic regression. Following standard practice, only variables that were significant in the bivariate analysis were included in the multivariate model to avoid overfitting. We have clarified this rationale in the revised manuscript, which can be found in lines 260-261.

Comment 8: The discussion needs to start from “The study found a 6.6% prevalence of ….”. Please delete the sentences about this, as they are unnecessary and redundant.

Authors Response: We appreciate this valuable comment and have agreed to revise the discussion as recommended. The prior sentences have been deleted, and the discussion now starts with 'The study found a 6.6% prevalence of ...'.

Comment 9: Under “strengths and limitations”, please start by stating the strengths of the study first; then write the limitations.

Authors Response: Thank you for your feedback! We have modified the section by stating the strengths of the study first, followed by the limitations.

Comment 10: One of the limitations that the authors did not discuss is the lack of data on BMI (or other measures of obesity), on dietary habits, diabetes, and on daily exercise.

Authors Response: Thank you for your valuable input. We have incorporated your suggestion in the revised manuscript (Lines 449-451)

Comment 10: Authors stated that this sampled population may genuinely represent the Tanzania population. I do not think this is correct, given the sample covers only women 15-49 years old, with mean age 29.3 years. The sample seems to be from a younger population, and we know that HBP is more prevalent in older people (e.g., 50+ years). In addition, this sample covers only women. There is evidence that sex is associated with HBP. Thus, this sample cannot be a reprehensive sample of the Tanzania population. The authors should acknowledge this as a limitation.

Authors Response: We appreciate your insightful comment regarding the representativeness of our sampled population. We acknowledge that this limitation may impact the generalizability of our results. The manuscript has been modified accordingly (lines 447-453).

Reviewer #2 Comments to the Author

Comment 1: In the Introduction, please add a paragraph on why it is important to investigate the prevalence of high blood pressure among women of reproductive age.

Authors Response: Thank you for the comment. We have incorporated the suggestion into the revised manuscript, can be found in Lines 142–144.

Comment 2: Why you chose this age group and please provide a literature review of similar studies about prevalence and determinants of high blood pressure about women of reproductive age.

Authors Response: In the revised manuscript, we have provided relevant evidence from global and regional studies that support the rationale for investigating this age group, as well as similar studies that have explored this topic (Lines 111 to 130).

Comment 3: In table 1, you can delete the frequency and percent for answer “No” to the self-reported high BP. Presenting data for yes, can show the frequency and percentage for “no” answer as well. I have the same comment for table 2.

Authors Response: Thank you for your recommendation regarding the presentation of data in Tables 1 and 2. However, we respectfully disagree with this recommendation, as we believe that including both the frequency and percentage for the 'No' responses, alongside the 'Yes' responses, enhances the clarity and comprehensiveness of our findings. This approach allows for a more nuanced understanding of the self-reported high blood pressure outcomes, thereby facilitating better interpretation of the data within the context of our study. We kindly request your agreement with our approach.

Comment 4: I recommend you present the bivariate analysis results in table 1 and the results of multivariate regression analysis in table 2.

Authors Response: Thank you for your recommendation regarding the presentation of the bivariate analysis results in Table 1 and the multivariate regression analysis results in Table 2. However, the intention of Table 1 is to display the characteristics of the study participants, while Table 2 focuses on the logistic regression analysis. We believe that maintaining this structure does not alter the findings of our study and provides clarity to the reader. Again, we kindly ask for your agreement on this view.

Comment 5: I recommend comparing the result of your study with the result of studies that reported the prevalence of high blood pressure in the same age group of female patients in Tanzania. It will show the difference between prevalence based on self-reported HBP and health facility reported HBP.

Authors Response: We appreciate your recommendation to compare our study's results with those reporting the prevalence of health facility-reported high blood pressure (HBP) in the same age group of female patients in Tanzania. The recommendation has been incorporated into the revised manuscript and can be found on line 363-365.

Comment 6: I recommend discussing more on the trend of HBP prevalence among women till 2030 and connect your recommendations to the Sustainable Development Goals.

Authors Response: Thank you for your insightful suggestion. However, given that our study focuses on a specific dataset and cross-sectional design, projections for future trends are outside the scope of the current analysis. Nevertheless, we acknowledge the importance of linking our findings to global health objectives and have expanded our discussion to better align our recommendations with the SDGs, particularly Goal 3, which aims to ensure healthy lives and promote well-being for all. This can be found on lines 431-435.

Reviewer #3 Comments to the Author

Comment 1: To further enhance the introduction, it would be helpful to include information on Tanzania's health indicators, such as the current prevalence of hypertension and the challenges within the healthcare system. Adding a brief overview of Tanzania's current efforts or policies to manage hypertension would also provide important context.

Authors Response: The recommendations have been incorporated into the revised manuscript. We have added information on the current prevalence of hypertension and the challenges within the healthcare system. These additions can be found in Lines 124–130

Comment 2: Concluding the introduction with a statement addressing the knowledge gap would strengthen the argument for the study's importance.

Authors Response: We appreciate this recommendation and have restructured the concluding statement in Lines 142–149.

Comment 3: In line 97, there is a small formatting issue as the space is missing between “reach1.56billion”.

Authors Response: Thank you for pointing out this formatting issue. We have corrected the spacing in line 97 to ensure it reads 'reach 1.56 billion' as intended

Comment 4: The study design (population-based cross-sectional) and the use of national survey data (TDHS 2022) are appropriate for the research question.

Authors Response: Thank you for your thoughtful assessment. We appreciate your recognition of the appropriateness of our study design and data sources for addressing the research question.

Comment 5: Sampling procedures, data collection methods, and the definition of the outcome variable (self-reported high blood pressure) are clearly described.

Authors Response: We are grateful for your positive feedback regarding the clarity of our sampling procedures and data collection methods. Your acknowledgment of these aspects enhances our confidence in the manuscript.

Comment 6: Statistical analysis is detailed, including the use of multivariable logistic regression and the presentation of adjusted odds ratios with confidence intervals.

Results:

Authors Response: Thank you for your commendation on the statistical analysis. We appreciate your recognition of the detail provided in our methodology, which is crucial for the interpretation of our results.

Comment 7: There are some sections where sentence structure can be simplified for better clarity and help convey the study’s key findings more effectively.

Authors Response: Thank you for your valuable feedback. We appreciate your suggestion to simplify sentence structures in certain sections. We have revised these areas throughout the manuscript to enhance clarity and effectively convey the study’s key findings.

Comment 8: It would benefit from integrating more context-specific information about Tanzania. For example, including information on how limited access to antenatal care services in rural Tanzania affects pregnancy-related hypertension risks would provide readers with a clearer understanding of the study's outcomes

Authors Response: Thank you for the suggestion. We have revised the discussion accordingly, integrating more context-specific information about Tanzania, particularly how limited access to antenatal care services in rural areas impacts pregnancy-related hypertension risks (lines 424-435)

Comment 9: The first two sentences in the discussion are repetitive as they already mentioned in the introduction. It may be helpful to paraphrase these sentences, start with a different sentence, or make them shorter.

Authors Response: Thank you for a valuable comment. This has also been raised by another reviewer. We have revised the discussion to start with 'The study found a 6.6% prevalence of ...', eliminating the previous sentences to avoid repetition.

The authors are indeed thankful for your time reviewing our manuscript and for your valuable input in shaping this research article to meet the journal’s standards and make it suitable for readers. We kindly invite you to review our revised manuscript and hope for its acceptance for publication.

We are looking forward hearing from you soon.

Best regards.

Nelson Musilanga, MD, MMed

Corresponding Author.

---

## [Decision Letter · Decision Letter 1]

19 Nov 2024

Burden and Determinants of Self-Reported High Blood Pressure Among Women of Reproductive Age In Tanzania: Evidence from 2022 Tanzania Demographic and Health Survey

PONE-D-24-35959R1

Dear Dr. Musilanga,

We’re pleased to inform you that your manuscript has been judged scientifically suitable for publication and will be formally accepted for publication once it meets all outstanding technical requirements.

Kind regards,

Muhammad Haroon Stanikzai

Academic Editor

PLOS ONE

Additional Editor Comments (optional):

A reviewer and myself found your responses and edits to the manuscript to be adequate and recommend acceptance for publication.

Reviewers' comments:

Reviewer's Responses to Questions

**Comments to the Author**

1. If the authors have adequately addressed your comments raised in a previous round of review and you feel that this manuscript is now acceptable for publication, you may indicate that here to bypass the “Comments to the Author” section, enter your conflict of interest statement in the “Confidential to Editor” section, and submit your "Accept" recommendation.

Reviewer #2: All comments have been addressed

2. Is the manuscript technically sound, and do the data support the conclusions?

Reviewer #2: Yes

3. Has the statistical analysis been performed appropriately and rigorously? 

Reviewer #2: Yes

4. Have the authors made all data underlying the findings in their manuscript fully available?

Reviewer #2: Yes

5. Is the manuscript presented in an intelligible fashion and written in standard English?

Reviewer #2: Yes

6. Review Comments to the Author

Reviewer #2: (No Response)

7. PLOS authors have the option to publish the peer review history of their article (what does this mean? ). If published, this will include your full peer review and any attached files.

**Do you want your identity to be public for this peer review?** For information about this choice, including consent withdrawal, please see our Privacy Policy .

Reviewer #2: **Yes: ** Narges Neyazi

---

## [Editor Report · Acceptance letter]

PONE-D-24-35959R1

PLOS ONE

Dear Dr. Musilanga,

I'm pleased to inform you that your manuscript has been deemed suitable for publication in PLOS ONE. Congratulations! Your manuscript is now being handed over to our production team.

Kind regards,

on behalf of

Dr. Muhammad Haroon Stanikzai

Academic Editor

PLOS ONE